# Evolution of an ancient protein function involved in organized multicellularity in animals

**Douglas P Anderson[1,2,3], Dustin S Whitney[4], Victor Hanson-Smith[1†], Arielle Woznica[5], William Campodonico-Burnett[2,3], Brian F Volkman[4], Nicole King[5], Joseph W Thornton[6,7*], Kenneth E Prehoda[2,3*]**

[1]Institute of Ecology and Evolution, University of Oregon, Eugene, United States; [2]Department of Chemistry and Biochemistry, University of Oregon, Eugene, United States; [3]Institute of Molecular Biology, University of Oregon, Eugene, United States; [4]Department of Biochemistry, Medical College of Wisconsin, Milwaukee, United States; [5]Department of Molecular and Cellular Biology, Howard Hughes Medical Institute, University of California, Berkeley, Berkeley, United States; [6]Department of Ecology and Evolution, University of Chicago, Chicago, United States; [7]Department of Human Genetics, University of Chicago, Chicago, United States

**\*For correspondence:** prehoda@molbio.uoregon.edu (KEP); joet1@uchicago.edu (JWT)

**Present address:** [†]Department of Microbiology/Immunology, University of California, San Francisco, United States

**Competing interests:** The authors declare that no competing interests exist.

**Abstract** To form and maintain organized tissues, multicellular organisms orient their mitotic spindles relative to neighboring cells. A molecular complex scaffolded by the GK protein-interaction domain (GK$_{PID}$) mediates spindle orientation in diverse animal taxa by linking microtubule motor proteins to a marker protein on the cell cortex localized by external cues. Here we illuminate how this complex evolved and commandeered control of spindle orientation from a more ancient mechanism. The complex was assembled through a series of molecular exploitation events, one of which – the evolution of GK$_{PID}$'s capacity to bind the cortical marker protein – can be recapitulated by reintroducing a single historical substitution into the reconstructed ancestral GK$_{PID}$. This change revealed and repurposed an ancient molecular surface that previously had a radically different function. We show how the physical simplicity of this binding interface enabled the evolution of a new protein function now essential to the biological complexity of many animals.

## Introduction

The evolution of organized multicellularity is one of the most important and least understood transitions in the history of life (*Grosberg and Strathmann, 2007*; *Maynard Smith and Szathmary, 1995*; *Bonner, 1998*; *King, 2004*). Multicellularity – defined as the differentiation and spatial arrangement of cell types into functioning tissues within an integrated organism – evolved independently in several eukaryotic lineages, using unique mechanisms each time to drive the cellular functions necessary for tissue organization (*Rokas, 2008*; *De Smet and Beeckman, 2011*; *Trillo and Nedelcu, 2015*). Comparative analyses have established that many protein families involved in cell adhesion, signal transduction, and cell differentiation in modern animals first appeared in the genomes of unicellular eukaryotes that were progenitors of animals (*King, 2003*; *Nichols et al., 2006*; *Richter and King, 2013*; *Rokas, 2008*). Virtually nothing is known, however, concerning the molecular mechanisms by which these proteins' functions evolved. These events happened in the deep past, so horizontal comparisons between extant species are often not sufficient to establish the historical changes in protein sequence, function, or biophysical properties that caused them. Vertical evolutionary analysis

using ancestral protein reconstruction – phylogenetic inference of ancestral sequences followed by gene synthesis, genetic manipulation, and experimental characterization – has proven to be an effective strategy for elucidating these questions (*Harms and Thornton, 2010*; *Harms and Thornton, 2013*). Here, we apply ancestral protein reconstruction to investigate the historical trajectory, timing, and mechanisms of evolution of a new protein function important to organized multicellularity in diverse animal phyla.

For dividing animal cells to generate and maintain organized tissues, the mitotic spindle must be oriented relative to the position of surrounding cells (*Morin and Bellaïche, 2011*; *Gillies and Cabernard, 2011*; *Lu and Johnston, 2013*; *Cabernard and Doe, 2009*; *Williams et al., 2011*). Cells that orient the spindle parallel to the epithelial plane, for example, expand the tissue; those that rotate it orthogonally to the plane escape the epithelium, as in epithelial-mesenchymal transitions during development (*Morin and Bellaïche, 2011*; *Gillies and Cabernard, 2011*; *Nakajima et al., 2013*). Experiments have identified a protein complex that mediates robust positioning of the mitotic spindle by using a scaffolding protein to link the spindle's astral microtubules to a molecular marker that is localized on the cell's cortex by external signals (*Figure 1A*) (*Lu and Johnston, 2013*; *Johnston et al., 2009*; *Siegrist and Doe, 2005*; *Siegrist, 2006*).

The complex and its functions have been most extensively studied in *Drosophila melanogaster* neuroblasts, but it plays a similar role in birds and mammals (*Saadaoui et al., 2014*) and in other cell types, such as several kinds of epithelium (*Nakajima et al., 2013*; *Saadaoui et al., 2014*; *Bergstralh et al., 2013*; *Bell et al., 2015*). In this complex, the scaffold is the GK protein interaction domain (GK$_{PID}$) of the protein Discs large (Dlg), which binds microtubule-associated motor proteins, such as the kinesin-3 family member KHC-73, and the Partner of Inscuteable protein (Pins in insects, LGN in vertebrates). In neuroblasts, the complex is localized relative to the position of adjacent cells by the interaction of a transmembrane receptor – which receives local extracellular signals – with Pins, which in turn recruits GK$_{PID}$, KHC-73, and the spindle microtubules (*Yoshiura et al., 2012*). In epithelia, in contrast, localization of the complex relative to surrounding cells appears to be mediated by Dlg itself; Pins is then recruited into the complex via interaction with Dlg's GK$_{PID}$. In addition to serving as a localized molecular mark in some cell types, Pins also brings other proteins to the complex, including Mud/Numa and its partners, which generate pulling forces and reinforce proper spindle orientation once it is established by the GK$_{PID}$ complex (*Lu and Johnston, 2013*; *Johnston et al., 2009*). Other molecules and pathways may be important in spindle orientation in other kinds of cells (*Morin and Bellaïche, 2011*; *Gillies and Cabernard, 2011*; *Lu and Johnston, 2013*), and further work is required to comprehensively assess the generality of the GK$_{PID}$ complex's role in spindle orientation across cell types and in the most basal animal lineages. Nevertheless, the fact that the GK$_{PID}$-mediated complex orients the mitotic spindle in multiple cell types in both protostomes and deuterostomes suggests an ancient and essential role in the biology of complex animals. Indeed, compromising Dlg's GK$_{PID}$ or other components of the Pins-Dlg-KHC-73 complex affects numerous tissues and cell types by causing impaired spindle orientation, tumorigenesis plasia, defects in cell polarity and differentiation, and developmental failures of tissue organization (*Nakajima et al., 2013*; *Johnston et al., 2009*; *Siegrist and Doe, 2005*; *Bergstralh et al., 2013*; *Yoshiura et al., 2012*; *Bilder, 2000*; *Woods, 1996*).

Little is known concerning the evolution of animal spindle orientation or the GK$_{PID}$-mediated complex in particular. Dlg is a member of a larger family of membrane-associated multidomain proteins, all of which contain a GK$_{PID}$ and form protein complexes important to cell adhesion, neural synapse organization, and other functions (*Funke et al., 2005*). The GK$_{PID}$ has been found only in animals, choanoflagellates, and Filasterea (*te Velthuis et al., 2007*; *de Mendoza et al., 2010*), but it is similar in both sequence and structure to the guanylate kinase (gk) enzymes, which are common to all life and regulate nucleotide homeostasis by catalyzing the transfer of phosphate groups from ATP to GMP (*Li et al., 1996*). These observations suggest that Dlg's GK$_{PID}$ may have evolved from an ancient gk enzyme (*Johnston et al., 2011*), but this hypothesis is untested and its evolutionary implications are unexplored. For example, it is not known when the GK$_{PID}$'s scaffolding functions first evolved, either in relation to the emergence of organized multicellularity or the origin of the other components of the spindle orientation complex; it is therefore unclear how the complex was assembled by evolution or what its role may have been in the emergence of spindle orientation and tissue

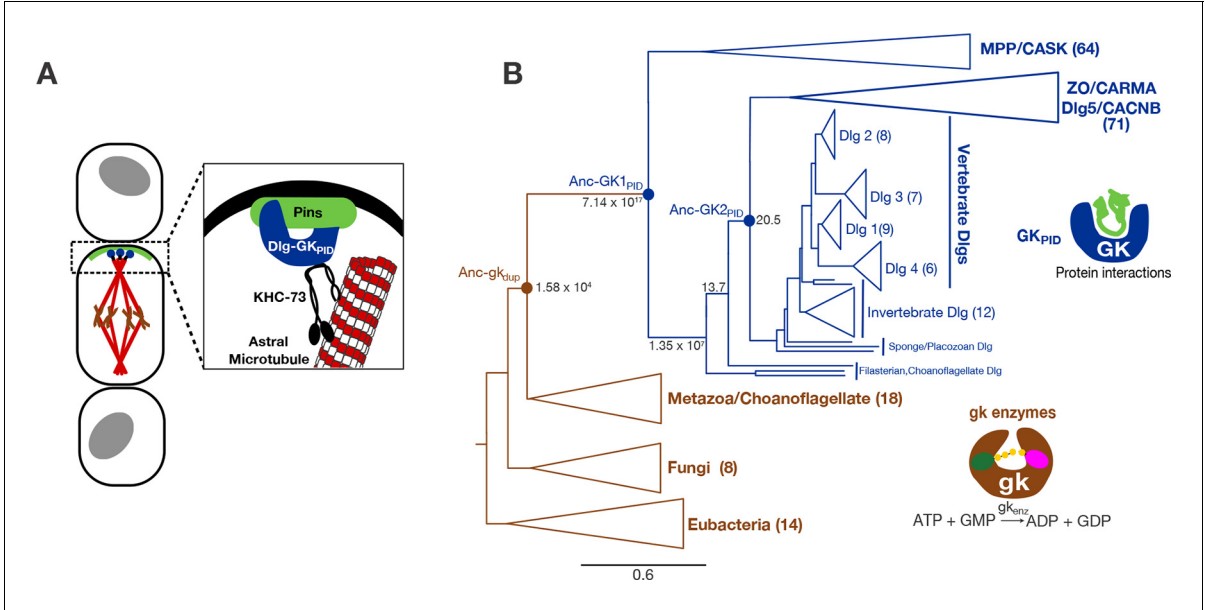

**Figure 1.** Function and phylogeny of the guanylate kinase (gk) and GK$_{PID}$ protein family. (**A**) The GK$_{PID}$ of the protein Discs-large (Dlg, blue) serves as a scaffold for spindle orientation by physically linking the localized cortical protein Pins (green) to astral microtubules (red) via the motor protein KHC-73 (black). (**B**) Reduced phylogeny of the protein family containing gk enzymes (brown) and protein-binding GK$_{PID}$s (blue). Parentheses show the number of sequences in each clade. Reconstructed proteins Anc-gk$_{dup}$ (the preduplication ancestor of gk enzymes and GK$_{PID}$s in animals/choanoflagellates), Anc-GK1$_{PID}$ and Anc-GK2$_{PID}$ (the GK$_{PID}$ in the common ancestor of animals and choanoflagellates, and of animals, respectively) are marked as circles with approximate likelihood ratio support. Scale bar indicates number of substitutions per site. For unreduced phylogeny, see *Figure 1—figure supplement 1*. Characteristics of the reconstructed sequences are found in *Figure 1—figure supplement 2*. For sequences analyzed, see *Figure 1—source data 1*. For sequences and posterior probabilities of amino acid states, see *Figure 1—source data 2*, *Figure 1—source data 3*.

The following source data and figure supplements are available for figure 1:

**Source data 1.** Species and identifiers for sequences used in alignment and phylogenetic analysis.

**Source data 2.** Posterior probability distribution of ancestral states for Ancgk$_{dup}$.

**Source data 3.** Posterior probability distribution of ancestral states for AncGK1$_{PID}$.

**Figure supplement 1.** Complete phylogeny of 224 guanylate kinase enzyme and GK$_{PID}$s.

**Figure supplement 2.** Sequence characteristics of maximum likelihood reconstructions of Anc-gk$_{dup}$ and Anc-GK1$_{PID}$.

organization. Some work has been done to identify amino acids that contribute to the functional differences between gk enzymes and GK$_{PID}$s in present-day organisms (*Johnston et al., 2011*), but it is unknown whether those residues are historically relevant to the evolution of GK$_{PID}$'s functions or whether they provide a necessary and sufficient explanation in the context of the ancestral protein to explain the emergence of the protein's spindle-orienting functions.

How an ancient gk enzyme gave rise to a protein domain with scaffolding/spindle orientation functions is also a striking model for the evolution of novel molecular functions. Most studies of protein evolution to date have focused on relatively subtle shifts in function, such as changes in relative ligand preference, in allosteric regulation, or in quantitative measures of activity (*Harms and Thornton, 2013*). GK$_{PID}$s and gk enzymes, however, have entirely different biochemical functions – specific protein binding and catalysis of a nucleotide substrate – suggesting the evolution of an entirely new function. Virtually nothing is known concerning the mechanisms and dynamics by which fundamentally novel protein functions evolve.

We therefore used ancestral protein reconstruction to reconstruct the sequences of ancestral members of the protein family that contains the gk enzymes and GK$_{PID}$s. This strategy allowed us to

trace their functional evolution through time and dissect the genetic and biophysical mechanisms that mediated the evolution of the GK$_{PID}$'s new functions. To understand the context in which GK$_{PID}$s evolved, we also sought insight into the evolution of other components of the spindle orientation machinery – and spindle orientation itself – by examining their presence in the single-celled eukaryotes most closely related to animals.

## Results and discussion

### Phylogeny of the gk/GK$_{PID}$ family

Understanding the historical process of protein evolution begins with a phylogeny. We first assembled a sequence alignment of the gk enzyme/GK$_{PID}$ family by searching publicly available databases and genomes. We recovered gk enzymes from taxa across the tree of life, suggesting a universal distribution, as previously reported (*te Velthuis et al., 2007*; *de Mendoza et al., 2010*). In contrast, GK$_{PID}$s are present in all animal genomes analyzed as well as in their closest unicellular relatives, the choanoflagellates and Filasterea; however, GK$_{PID}$s are absent from the genomes of all sequenced fungi and all other eukaryotic and prokaryotic lineages analyzed. We aligned 224 broadly sampled amino acid sequences and inferred the phylogeny of the gk enzyme/GK$_{PID}$ family using maximum likelihood phylogenetics, rooted using the bacterial gk enzymes as an outgroup (*Figure 1B*, *Figure 1—figure supplement 1*).

All GK$_{PID}$s cluster together as a monophyletic group, with the gk enzymes forming a paraphyletic set of basal lineages. Within the GK$_{PID}$s, there are two major clades, one of which contains Dlg and closely related paralogs; the other contains other family members, which are involved in cell adhesion and numerous other processes. Choanoflagellate and Filasterean genomes each contain both a gk enzyme and a GK$_{PID}$, with the latter proteins occupying a well-supported basal position sister to the metazoan Dlg-containing clade. This topology indicates that the gene that came to code for GK$_{PID}$s was generated by duplication of an ancient gk enzyme before the last common ancestor of Filozoa (animals+choanoflagellates+Filasterea) and after the split of Filozoa from the lineage leading to fungi, which contain no GK$_{PID}$s. Further gene duplications within the animals produced the diverse proteins that now contain GK$_{PID}$s (*Figure 1B*, ref. [*de Mendoza et al., 2010*]).

### Ancient evolution of GK$_{PID}$'s spindle-orienting functions

To understand how and when the scaffolding and spindle-orienting functions of the GK$_{PID}$ evolved, we used maximum likelihood (ML) phylogenetics to reconstruct ancestral sequences at critical nodes on the tree. We focused on two key ancestral proteins: Anc-GK1$_{PID}$, which represents the single GK$_{PID}$s from which GK$_{PID}$s in all Filozoan taxa descend) and its progenitor, Anc-gk$_{dup}$, which existed just before the gene duplication that split the gk enzymes from the GK$_{PID}$s. Most sites in Anc-gk$_{dup}$ were reconstructed with high confidence (mean posterior probability per site 0.94, with only 20 ambiguously reconstructed sites, defined as having a second plausible reconstruction with PP > 0.20); Anc-GK1$_{PID}$ was reconstructed with lower confidence (mean PP = 0.77, and 51 ambiguous sites, see *Figure 1—figure supplement 2* and *Figure 1—source data 2 and 3*)

To determine when GK$_{PID}$'s functions evolved, we synthesized DNAs coding for the ancestral protein sequences, expressed and purified the proteins in cultured cells, and characterized their functions by 1) measuring guanylate kinase activity in vitro using a coupled enzyme assay, 2) assessing affinity for a labeled Pins peptide using a fluorescence anisotropy assay, and 3) characterizing spindle-orienting function using an assay of mitotic spindle geometry in cultured cells transfected with a GK domain of interest. In the latter assay, *Drosophila* S2 cells in which native Dlg is knocked down were transfected with Pins fused to the cell-adhesion protein Echinoid, which localizes a crescent of Pins to the area of contact between adjacent cells; if and only if a functional GK$_{PID}$ is cotransfected will the spindle align during mitosis at a right angle to the crescent, along the axis between the two cells (*Johnston et al., 2009*).

We found that Anc-gk$_{dup}$ is an active guanylate kinase enzyme, with a Michaelis constant (K$_M$) comparable to that of the human enzyme, albeit with a slower k$_{cat}$ (*Figure 2A*). It displays no measurable Pins binding and failed to orient the mitotic spindle in living cells (*Figure 2B–E*). These data indicate that enzyme activity is, as predicted, the ancestral function of the family; further, the

scaffolding functions associated with spindle orientation were not yet present, even in suboptimal form, when duplication of the gk enzyme gene gave rise to the locus leading to $GK_{PID}s$.

By the time of the Filozoan ancestor, however, the evolving $GK_{PID}$ had lost the ancestral enzyme activity entirely and gained de novo spindle-orienting functions. Specifically, we found that Anc-$GK1_{PID}$ has no detectable guanylate kinase activity, but it binds Pins with moderate affinity and is highly effective in orienting the mitotic spindle in cell culture (*Figure 2A,B,F*). We also reconstructed Anc-$GK2_{PID}$ – the more recent progenitor of all Dlg proteins in the ancestral animal – and found that it too orients the mitotic spindle and binds Pins with even higher affinity, suggesting a subsequent fine-tuning of Pins-binding capacity (*Figure 2—figure supplement 1*).

It is unlikely that the ML ancestral reconstruction is precisely correct at all sites, so it is important to determine whether our conclusions about the functions of Anc-$gk_{dup}$ and Anc-$GK1_{PID}$ are robust to uncertainty about their inferred sequences. We therefore constructed an alternative version of each ancestral protein, in which all plausible alternative amino acid states (defined as those with posterior probability > 0.20) were introduced at once. These 'Alt-All' sequences represent the far edge of the cloud of plausible ancestral sequences, and they contain more differences from the ML reconstruction than the expected number of errors in the ML sequence (*Figure 1—figure supplement 2*). They therefore represent a conservative test of functional robustness to statistical uncertainty about the ancestral sequence. When assayed experimentally, the alternative version of Anc-$gk_{dup}$, like the ML reconstruction, was an active gk enzyme that did not bind Pins, and the alternative version of Anc-$GK1_{PID}$ bound Pins, as did the ML sequence (*Figure 2—figure supplement 2*). These results indicate that both the inferred trajectory of functional evolution and the phylogenetic interval during which protein scaffolding activity first evolved are robust to statistical uncertainty about the precise ancestral sequences.

Taken together, these findings indicate that the capacity of the $GK_{PID}$ to bind Pins and orient the mitotic spindle arose well before the evolution of animals or multicellularity itself. Indeed, these functions arose even before the divergence of the choanoflagellate and filasterean lineages and before the subsequent gene duplication that gave rise to the modern subgroups of $GK_{PID}$-containing proteins, including Dlgs. The capacity of $GK_{PID}s$ to carry out these functions therefore evolved before the evolution of organized multicellularity itself, because filastereans do not form organized colonies (although cells from some species do aggregate [*Sebé-Pedrós et al., 2013*]). This result is consistent with the hypothesis that the emergence of the $GK_{PID}$'s derived functions in protein scaffolding contributed to the subsequent evolution of animal complexity.

## Evolution of the spindle orientation complex by molecular exploitation

The evolution of $GK_{PID}$'s peptide-binding functions could have conferred spindle orientation only if its binding partners were already present. To understand when the other key components of the spindle orientation complex evolved, we characterized the taxonomic distribution of orthologs of metazoan KHC-73 and Pins by searching protein sequence databases and then inferring the age of each protein by parsimony-based inference on the taxonomic tree of life.

We found that KHC-73 orthologs are clearly present in animals, choanoflagellates, and Filasterea; fungal genomes do not contain a convincing ortholog of KHC-73 but do contain a closely related paralogous member of the kinesin-3 family (*Figure 3*, *Figure 3—source data 1*). This observation indicates that the KHC-73 gene is as old as the Filozoan ancestor and originated during the same phylogenetic interval in which Anc-$GK1_{PID}$ evolved its novel functions.

Pins orthologs have a more recent origin. The genome of *Salpingoeca rosetta*, a choanoflagellate that forms spatially organized spherical colonies (*Dayel et al., 2011*), contains an ortholog of Pins, which is very similar in sequence and domain architecture to its metazoan orthologs (*Figure 3*, *Figure 3—source data 1*, *Figure 3—source data 2*). A Pins ortholog is also present in another choanoflagellate, *Monosiga brevicollis*, but none was detected in Fungi, Filasterea, or any lineages outside of choanoflagellates and animals. In Filasterea, for example, the most similar protein to *D. melanogaster* Pins has a different domain structure, has detectable sequence similarity in only one small portion of the protein, and, when used as a query in a reciprocal search of the *D. melanogaster* genome, returns a protein from an entirely different family as the best hit with significance score 10 orders of magnitude better than its match to Pins (*Figure 3—source data 1*). Indeed, a key domain of Pins – the GoLoco domain, a simple 23-amino acid sequence that mediates Pins' contact with membrane-associated G-proteins and is therefore crucial to its cortical localization (*Smith and*

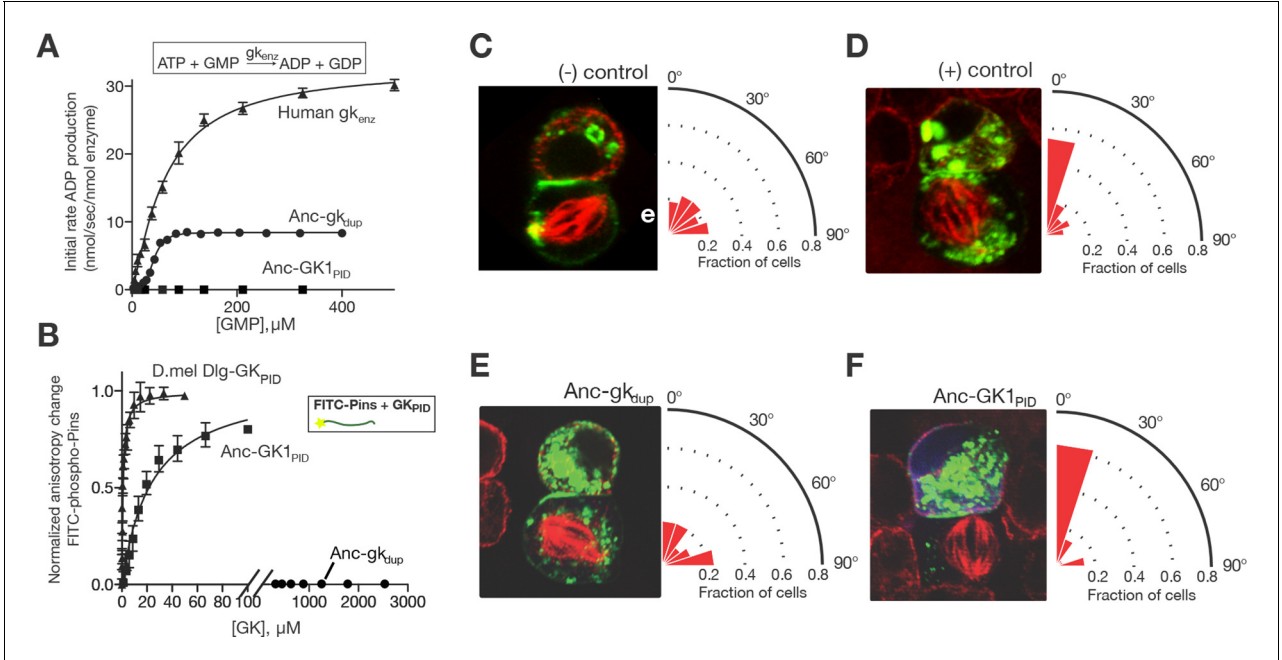

**Figure 2.** Evolution of a novel spindle-orientation function in ancestral GK$_{PID}$. (A) Anc-gk$_{dup}$ (circles) is an active nucleotide kinase in a coupled enzyme assay for the reaction shown; Anc-GK1$_{PID}$ (boxes) is inactive. Activity of the human gk enzyme (triangles) is shown for reference. Error bars show SEM for three replicates. (B) The more recent ancestral protein Anc-GK1$_{PID}$ (boxes) binds a 20 amino-acid peptide (see methods) from the Pins protein in a fluorescent anisotropy assay, but Anc-gk$_{dup}$ (cirlces) does not. Pins binding by the GK$_{PID}$ of the *Drosophila melanogaster* Dlg protein (triangles) is shown for reference. Error bars show SEM for three replicates. (C–F) Evolution of spindle orientation function as assayed in cultured S2 cells that do not express endogenous Dlg protein. Cells were transfected with a GK construct (C, –control: empty transfection vector; D, + control: GK$_{PID}$ from extant *Drosophila* Dlg) and scored for alignment of the mitotic spindle (red, tubulin, visualized immunocytochemically) relative to the Pins cortical crescent (green, a GFP-tagged Pins-Ecd fusion). In the example images for each experiment, two cells are shown, the bottom one of which is dividing. The angle of the mitotic spindle relative to a line bisecting the Pins crescent (from 0°, precisely aligned, to 90°) was recorded in many dividing cells; the radial histogram (right) shows the distribution of observed angles among all cells scored with a given genotype. Cells transfected with Anc-gk$_{dup}$ (E) do not display robust spindle orientation, but those transfected with Anc-GK1$_{PID}$ do (F). SEM: Standard error of the mean.

The following figure supplements are available for figure 2:

**Figure supplement 1.** Properties of ancestral protein AncGK2-PID.

**Figure supplement 2.** Robustness of functional inferences about ancestral proteins to uncertainty about the sequence reconstruction.

*Prehoda, 2011*) – exists only in animals and choanoflagellates. This observation establishes that Pins evolved before the split of choanoflagellates and animals and suggests that it evolved after a functional GK$_{PID}$ emerged in the Filozoan stem lineage.

The association of GK$_{PID}$ with Pins in the spindle orientation complex therefore evolved through a stepwise process of molecular exploitation (*Bridgham, 2006*) – a form of exaptation in which a newly evolved molecule recruits as a binding partner a more ancient molecule, which previously had different functions and a fortuitous but unrealized affinity for its new partner. In this case, GK$_{PID}$'s affinity for Pins apparently existed before the Pins protein came into existence. What could the functions of GK$_{PID}$ and KHC-73 have been before Pins evolved in the lineage leading to choanoflagellates and animals? GK$_{PID}$ and KHC-73 may have associated with each other to orient the mitotic spindle relative to a cellular mark other than Pins. Alternatively, GK$_{PID}$ may have functioned in other processes executed by its descendants, such as cell adhesion; KHC-73 also has numerous functions as an intracellular motor protein that are independent of spindle orientation (*Hanada, 2000*; *Horiguchi, 2006*). In either case, when a suitable form of Pins evolved considerably later, GK$_{PID}$ already had the capacity to bind it and thus form a scaffold for spindle orientation.

## Spindle orientation in choanoflagellates without GK-Pins association

The experiments reported above establish when $GK_{PID}$ evolved its capacity to bind extant animal Pins protein and to organize spindle orientation in extant animal cells; they do not reveal when a Pins protein that could bind $GK_{PID}$ first evolved. Pins is too poorly conserved for its ancestral form to be reconstructed with confidence. We therefore evaluated this question indirectly by investigating whether choanoflagellate $GK_{PID}$ binds choanoflagellate Pins, as expected if the association of Pins and $GK_{PID}$ and its role in spindle orientation predate the metazoan-choanoflagellate ancestor. We purified the $GK_{PID}$ of the *S. rosetta* Dlg-like protein and measured its affinity for the linker peptide of the Pins protein – the region to which $GK_{PID}$ binds (*Johnston et al., 2009*) – from *S. rosetta* and from metazoans.

We found that *S. rosetta* $GK_{PID}$ binds the *D. melanogaster* Pins linker peptide with moderate affinity (*Figure 4A*), corroborating our finding that $GK_{PID}$'s capacity to bind metazoan Pins originated before the common ancestor of animals and choanoflagellates. In contrast, *S. rosetta* $GK_{PID}$ does not detectably bind the *S. rosetta* Pins linker: in a simple pull-down assay using a glutathione-S-transferase/Pins fusion protein and His-tagged $GK_{PID}$, we detected no association of the two *S. rosetta* proteins under conditions in which the orthologous pair of proteins from *D. melanongaster* shows robust binding, even though both *S. rosetta* proteins are expressed and soluble (*Figure 4B*). That *S. rosetta* $GK_{PID}$ can bind the fruitfly's Pins but not its own suggests that Pins evolved its capacity to bind $GK_{PID}$ after animals diverged from choanoflagellates. We cannot rule out the less parsimonious possibilities that Pins lost an ancient capacity to bind $GK_{PID}$ in choanoflagellates or that some unique and unknown mode of association between $GK_{PID}$ and Pins operates in *S. rosetta*, such as requiring a bridging protein or some post-translational modification. Still, our experiments indicate that the mechanism of $GK_{PID}$'s association with Pins is not conserved between animals and choanoflagellates and may have evolved in the animal lineage after its divergence from choanoflagellates. Why *S. rosetta* $GK_{PID}$ can bind the *Drosophila* Pins linker but not its own is unclear; one possibility is that the surface of $GK_{PID}$ that fortuitously binds Pins is conserved in *S. rosetta* because it binds another structurally similar ligand, possibly an ancient one.

How does the history of the $GK_{PID}$-Pins complex relate to the evolution of spindle orientation? Spindle orientation itself is not a metazoan novelty; indeed, most eukaryotes – even unicellular ones – orient the mitotic spindle, but most appear to do so relative to the cell's internal structure alone rather than in response to cues from adjacent cells (*Wang et al., 2003*). We visualized the mitotic spindle in colonial *S. rosetta* using fluorescence microscopy and found that dividing *S. rosetta* cells appear to orient their mitotic spindles relative to the surface of the colony in a way that maintains the colony's spherical geometry (*Figure 4C*). If $GK_{PID}$ and Pins do not associate in choanoflagellates, how do colonial cells accomplish this? *S. rosetta* cells form colonies in a flagellum-out fashion, so orienting the spindle relative to the colony surface also entails orienting it at a right angle to the axis of the flagellum and, more generally, to the apical-basal (A-B) axis of the cell. We found that individual dividing *S. rosetta* cells that are not organized into colonies also orient their spindles relative to the flagellum and the A-B axis (*Figure 4D*). Because spindle orientation relative to the cellular axis occurs in both colonial and noncolonial *S. rosetta* cells, spindle orientation is likely to involve internal marks imposed by the cell's polarity rather than cues from neighboring cells.

Taken together, these findings suggest that spindle orientation mediated by the $GK_{PID}$-Pins complex in response to external cues replaced a more ancient mode of spindle orientation along the stem lineage leading to animals. This more ancient mechanism may have involved the flagellar basal bodies. The deeper choanoflagellate-metazoan ancestor almost certainly had flagella (*Nielsen, 2008*; *Buss, 1988*). Previous research indicates that during choanoflagellate mitosis, the flagellar basal body duplicates; the daughter bodies migrate symmetrically away from the cell's basal pole and then serve as microtubule organizing centers for assembling the spindle and orienting it perpendicular to the A-B axis (*Buss, 1988*; *Leadbeater, 2015*). A parsimonious hypothesis is therefore that spindle orientation in the choanoflagellate-animal ancestor was oriented relative to the A-B axis via the position of the flagellar basal bodies (*Buss, 1988*). This ancient mechanism would have been retained in choanoflagellates, allowing organized cell division in the context of flagellum-out spherical colonies. Along the branch leading to the metazoan ancestor, basal-body mediated orientation would have been replaced by the $GK_{PID}$-Pins association, providing a mechanism for the spindle to

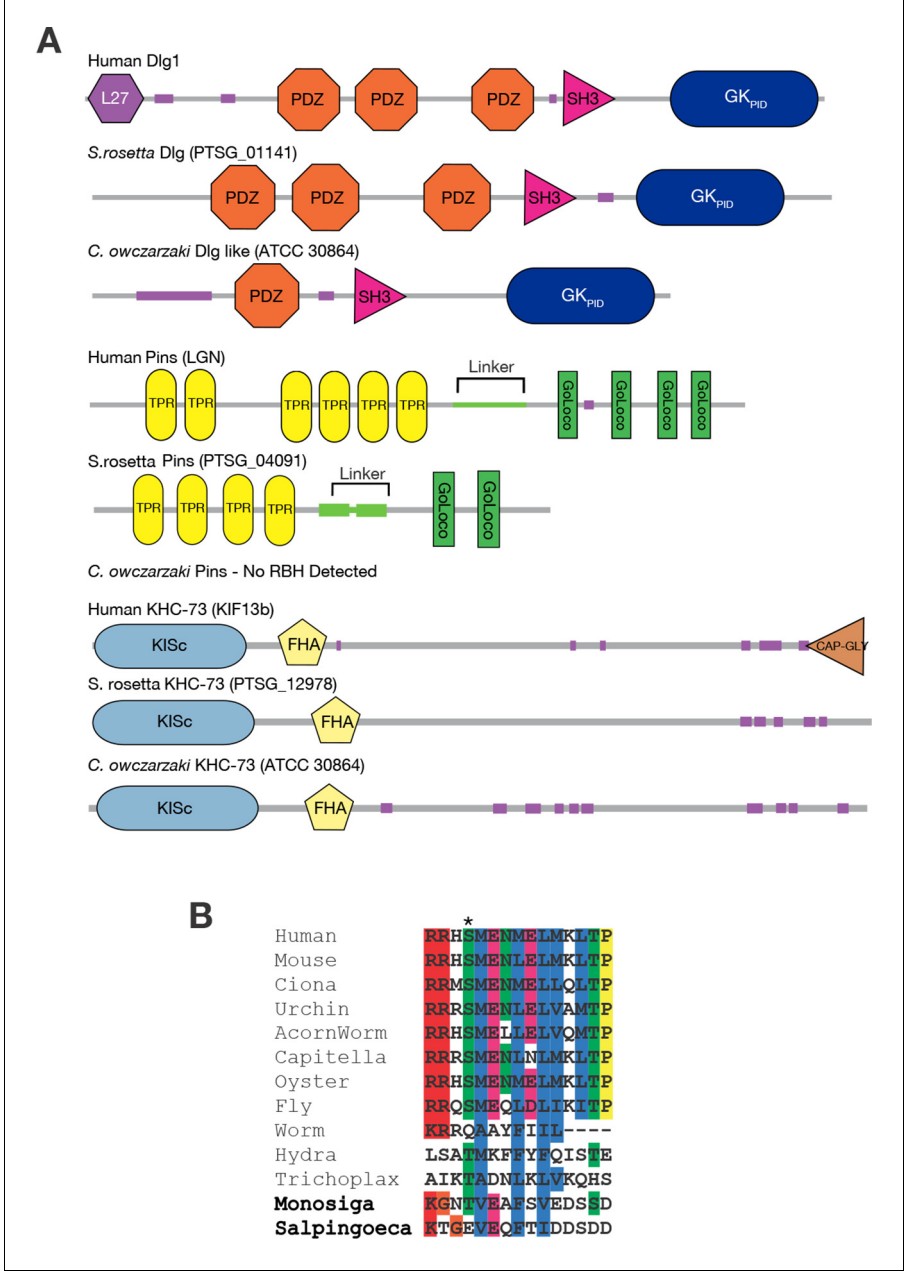

**Figure 3.** Taxonomic distribution of proteins in the spindle orientation complex. (**A**) The genomes of the choanoflagellate *Salpingoeca rosetta* and the filasterean *Capsaspora owczarzaki* contain orthologs of human Dlg, Pins, and KHC-73. The domain architecture of each protein is shown, as inferred using the SMART database. Each group of proteins are reciprocal best BLAST hits (RBH) to the human query protein shown. For details, see *Figure 3—source data 1*. (**B**) Aligned sequences from the linker portion of Pins (see panel A), which binds to Dlg. Colors highlight identical or biochemically conservative residues. Asterisk, phosphorylated or negatively charged residue 436, which in the *Drosophila melanogaster* Pins protein anchors Dlg binding. For complete species names and accessions, see *Figure 3—source data 2*.

The following source data is available for figure 3:

**Source data 1.** Results of reciprocal Blast search of metazoan and nonmetazoan genomes for orthologs of GK$_{PID}$, Pins and KHC-73.
**Source data 2.** Identifiers and species for Pins sequences shown in *Figure 3B*.

be oriented relative to adjacent cells via an externally organized molecular mark as the flagellum was lost from many animal cell types.

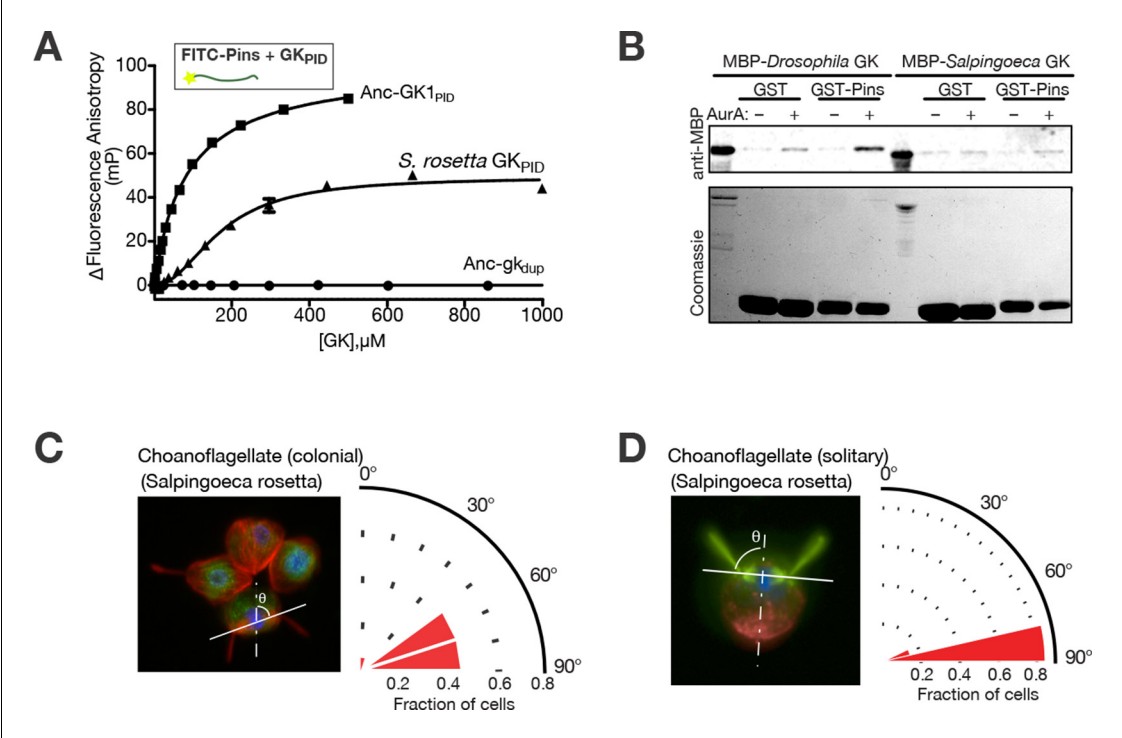

**Figure 4.** Spindle orientation and its molecular components in choanoflagellates. (A) Purified GK$_{PID}$ from the *Salpingoeca rosetta* Dlg ortholog (triangles) binds fluorescently labeled *Drosophila melanogaster* Pins. Binding by Anc-GK1$_{PID}$ (squares) and lack of binding by Anc-gk$_{dup}$ (circles) are shown for comparison. Error bars, SEM for three replicates. (B) Purified GK$_{PID}$ of *S. rosetta* does not interact with Pins of *S. rosetta;* under the same conditions, the *D. melanogaster* orthologs of these proteins do bind. A GST-fusion of each Pins linker regions (visible on the coomassie stained gel) was incubated with an MBP-fusion of each species' GK$_{PID}$ and detected by anti-MBP western blot. As previously found, binding of *D. melanogaster* Pins requires phosphorylation by Aurora A kinase; *S. rosetta* GK$_{PID}$ does not bind its own Pins whether or not the kinase is present. (C, D) Spindle orientation in colonial ( panel C, n = 12) and noncolonial (panel D, n = 7) *S. rosetta*. For each condition, the image shows one representative colony; the cell at bottom center is mitotic, as evidenced by condensed DNA (blue, DAPI) without a defined nuclear envelope (green, visualized using anti-nuclear pore complex). Red, mitotic spindle visualized using anti-tubulin. In colonial cells, the angle of the mitotic spindle (solid white line) was measured relative to a line perpendicular to the plane of the colony extending through the colony's center (dashed line). The histogram shows the distribution of spindle angles among all dividing *S. rosetta* cells measured, with 90° representing perfect alignment relative to the colony ring. In noncolonial cells, the spindle angle was measured relative to a line extending from the midpoint between the bases of the two duplicate flagella to the cell body's centroid (dotted line). An angle of 90° represents perfect alignment relative to the flagella. SEM: Standard error of the mean.

## A simple genetic basis for the evolution of GK$_{PID}$'s spindle-orienting functions

We next sought to identify the genetic mechanism by which the GK$_{PID}$ evolved its capacity to bind Pins and thus serve as a localized scaffold for spindle orientation. The potential candidate mutations are those that occurred during the phylogenetic interval between Anc-gk$_{dup}$ and Anc-GK1$_{PID}$ – the same branch on which the transition to GK$_{PID}$ function took place. Seventy-one amino acid replacements occurred along this branch, making identification of the functionally important changes an apparent challenge.

To identify potentially causal substitutions from this large set of candidates, we used both structure-function information and the phylogenetic pattern of sequence conservation/divergence within and between gk enzymes and GK$_{PID}$s. Extant gk enzymes contain two nucleotide binding lobes connected by a flexible hinge region around a central catalytic core (**Figure 5A**). In crystal structures of the gk enzyme in the absence of nucleotide substrate, the binding lobes are separated from each other in an open conformation (**Blaszczyk et al., 2001**). Upon nucleotide binding, the lobes move inward and occupy a closed conformation, bringing GMP and its co-substrate ATP together and allowing catalysis to occur (**Sekulic et al., 2002**). In contrast, the GK$_{PID}$ remains constitutively in the open conformation, and Pins – which is considerably larger than the enzyme's nucleotide ligands –

binds to the exposed surface of the guanylate-binding lobe (*Figure 5A*) (*Johnston et al., 2012*). We reasoned that the substitutions that caused the functional transition from enzyme to scaffold might have affected residues in the GMP/Pins binding interface or, alternatively, in the hinge that determines the orientation of the lobes relative to each other, thus affecting the size, geometry, or accessibility of the ligand-binding cleft.

Of the amino acid changes that occurred in these regions of the protein during the interval between Anc-gk$_{dup}$ and Anc-GK1$_{PID}$, only five are conserved among descendant GK$_{PID}$s (*Figure 5B, C*). To test these substitutions' functional importance, we introduced the derived states individually into Anc-gk$_{dup}$ and characterized their effects on guanylate kinase activity, Pins binding, and spindle orientation. Remarkably, we found that either of two amino acid changes in the hinge is sufficient to confer the protein-binding function. Substitution s36P, located where the hinge meets the binding lobe, virtually abolished the catalytic activity of Anc-gk$_{dup}$ and established moderate-affinity Pins binding (*Figure 6A,B*; lower and upper case residue symbols denote ancestral and derived states, respectively). In cultured cells, this single substitution also gave Anc-gk$_{dup}$ the capacity to robustly mediate spindle orientation (*Figure 6C*). Substitution f33S, also in the hinge, had largely similar effects, conferring Pins binding and decreasing – but not abolishing – enzyme activity; the affinity of Anc-gk$_{dup}$+f33S is almost identical to that of AncGK$_{dup}$+s36P, suggesting that f33S may also confer spindle orientation capacity, although this possibility was not directly tested. Combining f33S with s36P did not further shift the protein's function beyond that caused by either change alone (*Figure 6A*). In contrast, the three substitutions in the binding interface (s34C, a73G and f75Y) caused minor reductions in enzyme activity but did not confer even moderate Pins-binding (*Figure 6—figure supplement 1*).

The genetic basis for the evolutionary origin of the GK$_{PID}$'s spindle-orienting functions therefore appears to have been very simple. A single substitution – either s36P or f33S – was sufficient to recapitulate the evolution of the new functions that emerged during the evolution of the GK$_{PID}$ and to completely or partially abolish the ancestral function. The historical order of these amino acid changes is unknown. If s36P occurred first, it would have conferred on the protein all the major aspects of the functional transition, and f33S would have been subsequently inconsequential. If f33S occurred first, however, our data suggest that it would have generated a functionally hybrid intermediate that both bound Pins and retained some enzyme activity; s36P would then have completed the functional transition by abolishing the residual enzyme activity.

The two causal historical substitutions that we identified are both reconstructed without any statistical ambiguity at the relevant nodes. They are not identical to the set of mutations found in a previous study to be important to GK$_{PID}$ functions based on a horizontal comparison and mutagenesis of extant proteins from *D. melanogaster* and *S. cerevisiae* (*Johnston et al., 2011*). Replacement s36P was found in both studies, but the historical replacement f33S was not identified in the horizontal comparison. The other mutation identified in the horizontal comparison (S34P) never occurred during history, and we found that the historical amino acid replacement that did take place at this site (s34C) does not confer GK$_{PID}$-like functions (*Figure 6—figure supplement 1*).

## Repurposing an ancient binding interface

How could single amino acid changes have caused an entirely new function to evolve? Residue 33 does not contact the ligand, and residue 36 makes only a minor and apparently nonspecific contact to Pins at one end of the binding site (*Figure 7—figure supplement 1*). Overall, the residues that compose the Pins-binding surface are almost entirely conserved from Anc-gk$_{dup}$ to Anc-GK1$_{PID}$ (*Figures 5B*, *7A,B*), and the backbone structure of the binding lobe is almost identical between the crystal structures of extant GK$_{PID}$s, gk apo-enzymes, and a gk apo-enzyme containing mutation S36P (*Figure 7B*, see also ref. *Johnston, et al., 2011*). Thus, the Pins binding surface appears to have been present, even before the interaction with Pins itself evolved.

Why would a latent Pins binding surface have been present? Homology models of the ancestral proteins, as well as the structures of extant family members, indicate that the key portion of the Pins-binding surface of GK$_{PID}$ was derived without significant modification from the ancient surface that gk enzymes use to bind GMP (*Figure 7A,B*, *Figure 7—figure supplement 1*). This GMP-binding site could be repurposed for binding Pins because the two ligands – one a nucleotide, the other a peptide – share a key structural feature: a negatively charged head flanked by a small hydrophobic region (*Figure 7A,B*, ref. *Johnston, et al., 2012*). Specifically, the phosphate group at the head of

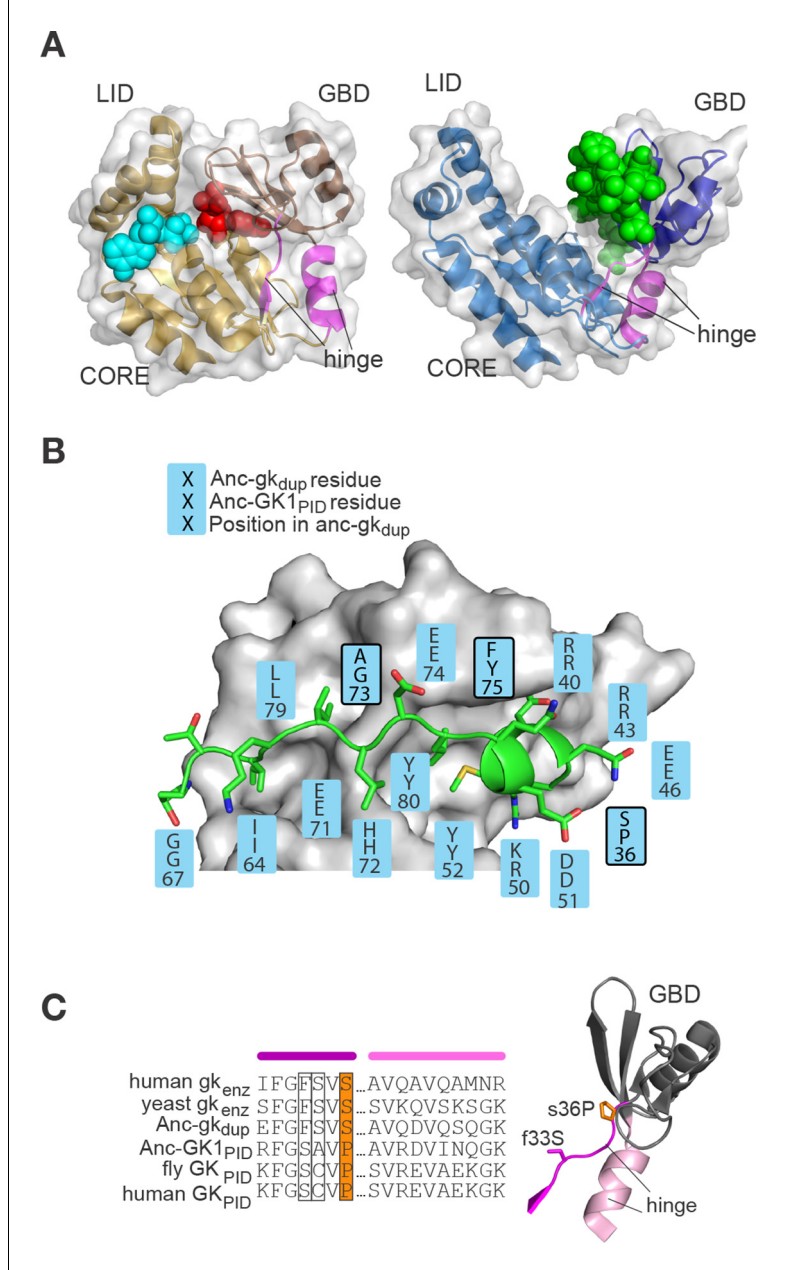

**Figure 5.** Evolution of the binding interface and hinge during the evolution of AncGK$_{PID}$ spindle orientation functions. (**A**) All gk/GK$_{PID}$ family members share a common structural architecture, comprising a catalytic core, two binding lobes (the GMP-binding domain, GBD, shown in dark hue, and the ATP-binding lid), and a flexible hinge region, which connects the GBD to the core and comprises two segments of contiguous residues (magenta). Left: in gk enzymes bound to GMP (red spheres), the lobes adopt a closed conformation, bringing GMP and ATP (cyan spheres) adjacent to each other in the core. Right: the GK$_{PID}$ has an open conformation, in which Pins (green spheres) binds to the surface of the GBD in the cleft between the two binding lobes. Structures shown are mouse gk enzyme (brown, PDB 1LVG) and the GK$_{PID}$ from rat Dlg1 (blue, 3UAT). (**B**) Most residues in Anc-GK1$_{PID}$ that bind Pins (blue boxes) are unchanged from the ancestral state in Anc-gk$_{dup}$. White surface, *D. melanogaster* Dlg GK1$_{PID}$ (3TVT). Green, Pins peptide. Ancestral and derived amino acid states are shown; residues with historical amino acid replacements between the two ancestral proteins are outlined. (**C**) In the hinge region, two historical substitutions (outlined and colored, with side-chains shown as sticks) were conserved in the ancestral state in extant enzymes and in a different state in extant GK$_{PID}$s. Colored bars above the sequence indicate position in the protein structure (right). Hinge segments are shown in pink and the GMP-binding lobe in gray.

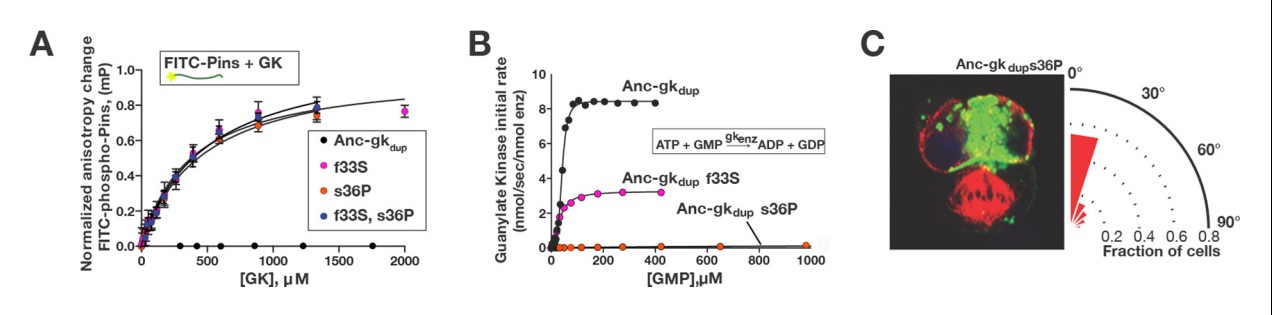

**Figure 6.** A simple genetic basis for the evolution of spindle orientation functions in the ancestral GK_PID. (A,B) Introducing either of two historical substitutions s36P or f33S into Anc-gk_dup confers Pins binding and reduces ancestral guanylate kinase activity. Error bars are SEM of 3 replicates. Combining s36P and f33S does not further affect the derived functions beyond the effects of the single substitutions. (C) Introducing s36P into Anc-gk_dup is sufficient to confer the full capacity to drive orientation of the mitotic spindle (compare *Figure 1*). As shown in *Figure 6—figure supplement 1*, historical substitutions in the binding interface do not recapitulate the evolution of Pins binding and the loss of gk activity.

The following figure supplement is available for figure 6:

**Figure supplement 1.** Introducing other historical substitutions into AncGK_du does not confer GK_PID-like function.

GMP is anchored to the enzyme with hydrogen bonds to four clustered residues that form a positively charged pocket (R40, R43, Y80, Y82); in the GK_PID, these same residues form hydrogen bonds to the phosphate group of Pins' phospho-serine 436. At the other end of GMP, the hydrophobic guanosine ring occupies a hydrophobic groove in the enzyme, and the same groove on the GK_PID binds to a hydrophobic methionine side chain on Pins. Additional interactions beyond this common interface are required for Pins binding (*Zhu et al., 2011*), but the physical interactions are relatively simple, including a series of small hydrophobic patches that bind to hydrophobic side chains on Pins and just two additional hydrogen bonds, both from backbone atoms on Pins to polar residues on the GK_PID surface that were solvent-exposed in the enzyme. Thus, the fortuitous similarity between GMP and one portion of Pins, along with the fortuitous arrangement of hydrophobic patches near the ancestral enzyme's GMP-binding site, made it possible for GK_PID to evolve Pins binding by a very minor genetic modification.

If the binding surface for Pins was already present in latent form in the Anc-gk_dup enzyme, how did s36P or f33S confer Pins binding? Several lines of evidence suggest that these substitutions altered the protein's dynamics and/or increased the relative occupancy of a conformation in which the latent binding site is exposed for peptide binding. First, the hinge where these two residues are located (*Figures 5C, 7B*) is known to mediate the dynamic opening/closing of the binding lobes relative to each other (*Blaszczyk et al., 2001*). Second, the degree to which the domain is open or closed appears to be essential for function. In gk enzymes, closing is critical for catalysis, because it brings the nucleotide substrates close together in the protected active site (*Bhabha et al., 2013*); in GK_PIDs, however, open conformations appear to be required for Pins binding, because the Pins peptide is significantly larger and is therefore predicted to sterically clash with the GK_PID when the lobes are close together. Third, substituting a proline at position 36 could restrict backbone dihedral angles in the hinge, possibly altering the dynamics of the hinge and/or the distribution of conformations it occupies. Replacing the bulky phenylalanine at position 36 could also change backbone dynamics, altering the occupancy of conformations in this region. Fourth, introducing a proline at residue 36 into extant gk enzymes has been shown to impede the GMP-induced closing motion, abolish enzyme activity, and to confer Pins binding (*Johnston et al., 2011*). Because the effects of mutation s36P on the function of the ancestral gk enzyme are nearly identical to those it has on the extant enzyme, it is likely that similar biophysical mechanisms pertain in the two proteins. Fifth, engineering yet a third mutation in the hinge – a serine-to-proline replacement at site 34 – into extant gk enzymes has been previously found to reduce enzyme activity, confer spindle orientation, and impede conformational closing, further supporting a causal link between these phenomena (*Johnston et al., 2011*).

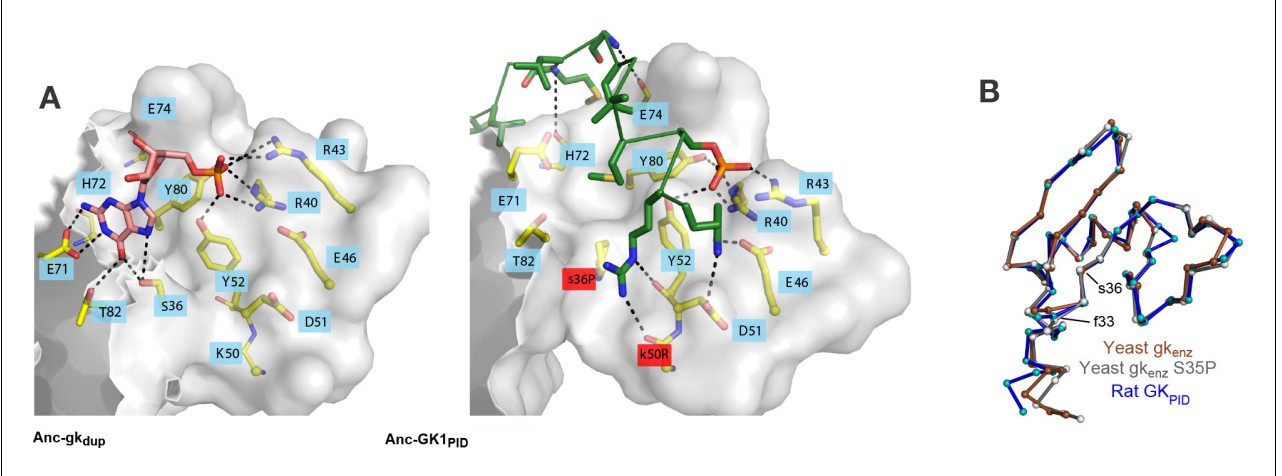

**Figure 7.** Evolution of GK$_{PID}$'s new function by unveiling a latent protein-binding site. (**A**) The binding surface for Pins in GK$_{PID}$s is derived from the GMP-binding surface of gk enzymes. Homology models of Anc-gk$_{dup}$ (left) and Anc-GK1$_{PID}$ (right) are shown as white surface, with all side chains that contact either GMP or Pins as yellow sticks. Pink sticks show GMP; green ribbon shows Pins backbone, with the side chains of all Pins residues that contact the GK protein shown as sticks. The phosphate group on GMP and on Pins residue 436 are shown as orange and red sticks. Black dotted lines, protein-ligand hydrogen bonds. In the AncGK1$_{PID}$ structure , substitutions at sites in the binding interface are shaded red, including key substitution s36P. The binding modes of extant gk enzymes and GK$_{PID}$s are similar and support the same conclusions (see *Figure 7—figure supplement 1*). (**B**) The structure of the hinge and GMP/Pins-binding lobes is conserved between the Pins-bound GK$_{PID}$ (blue, rat Dlg, 3UAT), the apo-gk enzyme (brown, *S. cerevisiae* guanylate kinase 1EX6), and the apo-gk-s36P mutant (gray, 4F4J), all in the open conformation.

The following figure supplement is available for figure 7:

**Figure supplement 1.** Structural context of key historical mutations.

We therefore propose that the historical hinge substitutions s36P or f33S inhibited enzyme activity and conferred the novel scaffolding function on GK$_{PID}$ by restricting the ancestral enzyme's dynamic hinge motions and/or changing its occupancy of conformations that have high affinity for Pins. This scenario is consistent with recent findings that changes in conformational occupancy may play an important role in the evolution of protein function (*Harms et al., 2013*; *Tokuriki and Tawfik, 2009*). Subsequent substitutions apparently fine-tuned Pins-binding affinity, yielding the higher-affinity Anc-GK1$_{PID}$ and Anc-GK2$_{PID}$. In addition to its likely effects on conformational occupancy and/or dynamics, Pro36 makes van der Waals contact to the Pins ligand (*Figure 7—figure supplement 1*), so it is possible that s36P may also have contributed to optimizing the latent Pins binding interface itself; in contrast, f33S does not contact the ligand and is therefore unlikely to have directly affected the interface.

## Simplicity and complexity in cellular evolution

A central issue in evolutionary biology is how complex systems originate through the action of mutation, drift, and natural selection. Tissue organization, spindle orientation, and the GK$_{PID}$ complex itself are all examples of complexity, defined as the integrated functioning of a system made up of differentiated, interacting parts. The GK$_{PID}$ complex can orient the mitotic spindle because of specific interactions between its component molecules and with other molecules in the cell and its local environment. In turn, the cellular phomonenon mediated by this complex – regular orientation of the plane of cell division relative adjacent cells – allows the development and maintenance of organized, differentiated tissues, and this phenomenon in turn makes possible a higher level of biological complexity– the multicellular organism – from a collection of individual cells. Understanding the evolution of complexity at the molecular level can therefore help to illuminate the evolution of macroscopic complexity, including functions that are now crucial to animal biology per se.

Our work indicates that the GK$_{PID}$ complex was assembled stepwise through a process of molecular exploitation, in which old molecules with one function are recruited into a functional binding interaction with a newly evolved molecule. In this case, the GK$_{PID}$, a duplicate of an ancient enzyme

with an essential metabolic role in all life forms, already had the fortuitous capacity to bind the Pins protein, even before the latter protein appeared or subsequently acquired its relatively simple GK<sub>PID</sub>-binding linker motif (*Figure 8*). Once Pins did evolve this linker — along with its GoLoco motif, which interacts with G-protein complexes, which are also ancient (*de Mendoza et al., 2014*)— then a mechanism would have been assembled that could bring to specific locations on the cell cortex the GK<sub>PID</sub> and other proteins associated directly or indirectly with it, such as KHC-73 and astral microtubules, thus enabling externally-cued spindle orientation.

Our analyses do not establish a complete history of the spindle orientation complex. Many key steps remain to be reconstructed, including how and when the interaction between GK<sub>PID</sub> and KHC-73 evolved, the mechanisms by which Pins' acquired its linker and GoLoco sequences, and the relationship of these components to other molecular complexes and pathways involved in animal spindle orientation. Despite these knowledge gaps, our observations establish a broad overview of the history of the GK<sub>PID</sub> complex, provide a detailed mechanistic reconstruction of a key event, and point to the importance of reusing molecules – and specific surfaces within them – for fortuitous new purposes that have the potential to become biologically essential.

Other reported cases of molecular exploitation have involved recruiting new binding partners that are subtle variants of its parent's ligand – such as steroid hormones with a modified functional group at a key position or a minor change in the hormone's structure (*Bridgham, 2006*; *Harms et al., 2013*; *Carroll et al., 2008*). In contrast, GK<sub>PID</sub> acquired an entirely new function – from enzyme ancestor to protein-binding scaffold – and affinity for an entirely different class of macromolecule – from nucleotide to peptide, and it did so through as little as one historical change in amino acid sequence.

The genetic simplicity of the evolutionary change in GK<sub>PID</sub> function is underscored by the fact that we found not one but two historical amino acid replacements from the relevant phylogenetic interval, either of which is sufficient to confer the GK<sub>PID</sub>'s derived functions on the ancestral enzyme. This finding indicates that GK acquired its new protein-binding function through a relatively simple, high-probability genetic path, rather than a long trajectory that required many specific mutations before the new function could be established.

GK<sub>PID</sub>'s dramatic evolutionary transition in function could take place through such a simple genetic mechanism because of its biophysical architecture. The gk enzyme's simple binding site for GMP can also be occupied by a simple two-residue motif on the Pins peptide, which fortuitously has similar surface properties. In addition, a series of small hydrophobic patches, which happen to be adjacent, was available to bind the hydrophobic portion of the Pins peptide and increase affinity. All that was required to confer the protein's new function was a single mutation that revealed this molecular surface, apparently by changing the protein's conformational flexibility. In this way, the physical simplicity of an interaction between ancient molecules set the stage for the easy evolution of a novel molecular complex and, in turn, a cellular function that now plays an important role in the complex biology of multicellular animals.

## Materials and methods

### Phylogenetics, ancestral protein reconstruction, homology modeling, and ortholog identification

Annotated protein sequences of 224 guanylate kinases and GK<sub>PID</sub>s were downloaded from UniPROTKB/TrEMBL, GenBank, the JGI genome browser, and Ensemble databases. Amino acid sequences were aligned using MUSCLE (*Edgar, 2004*), followed by manual curation and removal of lineage-specific indels. For species and accessions used, see *Figure 1—source data 1*. Guanylate kinase sequences were trimmed to include only the active gk domain predicted by the Simple Modular Architecture Research Tool (SMART) (*Schultz et al., 1998*).

The phylogeny was inferred by ML using PhyML v2.4.5 (*Guindon et al., 2010*) and the WAG model with gamma-distributed rate variation and empirical state frequencies, which was selected using ProtTest software and the AIC criterion. Statistical support for each node was evaluated by obtaining the approximate likelihood ratio (the likelihood of the best tree with the node divided by the likelihood of the best tree without the node) and the chi-squared confidence statistic derived from that ratio (*Anisimova et al., 2011*). Ancestral protein sequences and their posterior probability

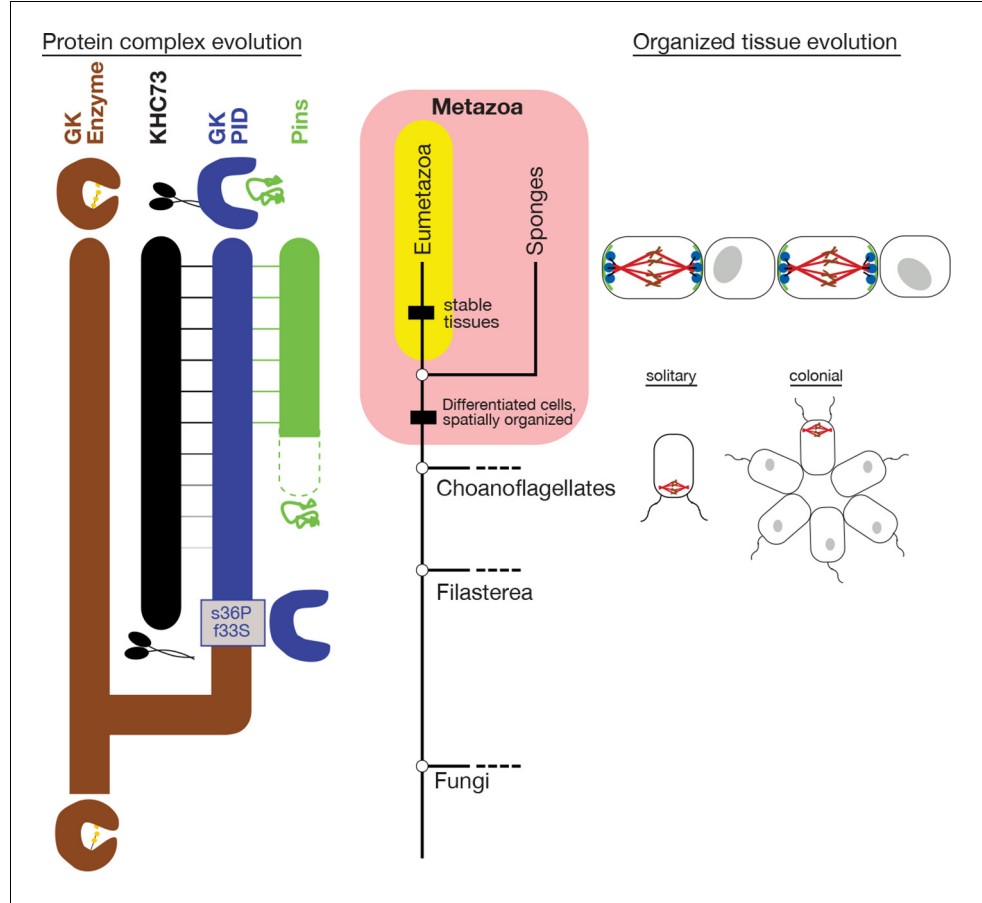

**Figure 8.** Historical evolution of GK$_{PID}$-mediated spindle orientation complex. The center portion shows the phylogeny of Metazoa and closely related taxa. The origin of cell differentiation and spatially organized tissues is marked. The left portion shows major events in the evolution of the components of the spindle orientation complex reconstructed in this study. Duplication of an ancestral gk enzyme (brown) and the key mutations that led to the origin of a GK$_{PID}$ (blue) that could bind other molecules in the complex are shown relative to the phylogeny's time scale. The apparent date of origin of KHC-73 (black) and Pins (green) are also shown. Dotted green line shows the origin of Pins in a form not yet bound by GK$_{PID}$. Solid green line shows GK$_{PID}$-binding form. Horizontal lines indicate binding between proteins. The right portion shows a schematic of the spindle orientation machinery in metazoans, which allows orientation relative to external cues from nearby cells, as well as spindle orientation relative to the internal cell axis as marked by the flagella in both solitary and colonial choanoflagellates.

distributions were inferred by the ML/empirical Bayes method (*Yang et al., 1995*), assuming the ML phylogeny and the best-fit model, using PAML v3.13 and Lazarus software (*Hanson-Smith et al., 2010*). Average probabilities were calculated across all GK sites except those containing indels. Plausible alternative non-ML states were defined as those with posterior probability >0.20. Alternate ancestral sequences at each node were prepared containing the ML state at all unambiguously reconstructed sites and the state with the second highest posterior probability at all ambiguously reconstructed sites. Sequences of reconstructed proteins Anc-gk$_{dup}$, Anc-GK1$_{PID}$, and Anc-GK2$_{PID}$ were deposited in Genbank with identifiers KP068002, KP068003, and KP068004, respectively.

Homology modeling was performed with MODELLER software using the automodel single comparison class. 1GKY and 1EX6 were used as the template for Anc-gk$_{dup}$ and 3UAT for Anc-GK1$_{PID}$. For each protein, 10 homology models were prepared, and that with the lowest DOPE score was used as the best model (*Eswar et al., 2006*).

Choanoflagellate and filasterean genes orthologous to metazoan guanylate kinase, Dlg, Pins, and KHC-73 proteins were identified by using the NCBI BLAST tool using *Homo sapiens* or *Drosophila*

*melanogaster* protein sequences as queries. For each search, the top hit in the target species was verified with a reciprocal BLAST search against the query genome. Domain architecture similarity was assessed using the SMART domain recognition tool (*Schultz et al., 1998*).

## Protein expression

DNA sequences coding for reconstructed ancestral proteins and optimized for *E. coli* codon bias were synthesized (Genscript) and then inserted into pBH plasmid vector with a hexa-His tag for *E. coli* expression at 20°C. Protein purification was carried out using sequential NiNTA affinity, anion exchange, and size-exclusion chromatographies. All proteins eluted as predicted monomers from the size-exclusion column at purity >95% by Coomassie staining of an SDS–PAGE gel. Proteins were concentrated using Vivaspin concentrators (Sigma-Aldrich), flash frozen in liquid nitrogen, and stored at −80°C in buffer (20 mM Tris, pH 7.5, 150 mM NaCl, 1 mM DTT).

## Guanylate kinase activity assay

We used a coupled enzyme assay, as described previously (*Agarwal et al., 1978*), which quantifies release of ADP in the guanylate kinase-catalyzed reaction by coupling it to pyruvate kinase- and lactate dehydrogenase-catalyzed reactions and measuring the consequent oxidation of NADH by following the decrease in absorbance at 340 nm, which we measured on a Tecan Safire plate reader. Guanylate kinase enzyme was at 50–200 nM in assay buffer (100 mM Tris, pH 7.5, 100 mM KCl, 10 mM MgCl2, 1.5 mM sodium phosphoenolpyruvate, 300 mM NADH, 4 mM ATP, and 100 units pyruvate kinase and 100 units lactate dehydrogenase). Initial GMP concentrations ranged from 500 nM to 1 mM. The reaction was initiated by adding GMP and briefly mixing. Reactions were carried out at 30°C and measured 30 times at 15s intervals. Data were analyzed and plotted using GraphPad Prism assuming Michaelis-Menten kinetics. Reaction rates are plotted as initial rate of ADP production.

## Protein binding assays

Binding of *D. melanogaster* Pins was assayed by fluorescence anisotropy on a Tecan Sapphire plate reader equipped with automatic polarizers using default settings for anisotropy assays. A $NH_2$-terminal FITC-labeled peptide (GVRVRRQ(pS)MEQLDLIKITPD, Genscript) of the fly Pins-Linker peptide (0.25 µM) was incubated with increasing concentrations of GK protein in phospho-buffered saline solution with 1mM DTT. A one-site binding model was used to fit the data and infer binding affinity in Graphpad Prism. Attempts to measure affinity of the *S. rosetta* $GK_{PID}$ for the *S. rosetta* Pins linker using MBP pull-down experiments were unsuccessful under a variety of assay conditions due to loss of the immobilized MBP-Pins fusion protein upon addition of $GK_{PID}$. Ancestral Pins peptides were not reconstructed because Anc-gk$_{dup}$ and Anc-GK1$_{PID}$ both correspond to gene duplications rather than speciation events; no nodes precisely corresponding to the same genomes would exist on a Pins phylogeny. Our inferences are therefore limited to the capacity of ancestral Anc-gk$_{dup}$ and Anc-GK1$_{PID}$ to bind to Pins peptides found in extant animals.

For GST pull-downs, we expressed GST fused to the 20-residue Pins linker of *S. rosetta* (PRGSKT-GEVEQFTIDDSDD) or *D. melanogaster* (GVRVRRQSMEQLDLIKITPD) and MBP-tagged $GK_{PID}$ proteins from the same species. Proteins were expressed separately in BL21(DE3) *E. coli* cultures and purified with glutathione or amylose agarose, respectively. All four proteins expressed in soluble form. The GST-Pins fusion was left attached to the resin and incubated with MBP-tagged $GK_{PID}$, followed by three washes. Bound $GK_{PID}$ was eluted in SDS-PAGE loading buffer and visualized using Western electrophoresis and an anti-MBP antibody (Qiagen); unbound $GK_{PID}$ from the final wash was visualized by electrophoresis and Coommassie staining. Binding reactions were performed in the absence and presence of Aurora A kinase (Calbiochem), which is necessary for phosphorylation of the *D. melanogaster* Pins linker at residue S436 and subsequent $GK_{PID}$ binding (*Johnston et al., 2009*).

## Spindle orientation assays

Maintenance of S2 cells, construction of expression plasmids, and cell-adhesion/spindle orientation assays were performed as detailed previously (*Johnston et al., 2011*). S2 cells were transfected with constructs coding for a FLAG- or HA-tagged GK protein and for a Pins-GFP-Echinoid fusion protein,

using Effectene reagent (Qiagen, Germantown, MD) and 0.4–1 µg total DNA for 24–48 hr. Endogenous Dlg was knocked down using RNAi: transfected cells were incubated for 1 hr in serum-free media containing approximately 1 µg RNAi followed by 72 hr in normal growth media. Protein expression was induced by adding 500 µM CuSO4 for 24 hr. Cell-adhesion clustering was induced by constant rotation at approximately 175 rpm for 1–3 hr. Pins fusion protein was visualized by fluorescence (excitation 488 nm, emission 509 nm). Mitotic spindles were visualized using rat anti-tubulin (Abcam 1:500) and goat anti-rat conjugated to Alexa 555 (Life Technologies, 1:500). GK was visualized using mouse anti-FLAG or anti-HA (Sigma 1:500) and chicken anti-mouse:Alexa 647 (Life Technologies 1:200); histones were visualized using rabbit antiphospho-histone3 (Upstate 1:8000) and donkey anti-rabbit:Alexa 488 (Life Technologies, 1:500). For immunostaining, clustered cells were fixed in 4% paraformaldehyde for 20 min, washed, and incubated with primary antibodies overnight at 4°C in buffer (0.01% saponin plus 0.1% albumin diluted in phosphate-buffered saline). Slides were subsequently washed and fluorescently linked secondary antibodies were added for 2 hr at room temperature (RT). Finally, slides were again washed and mounted using Vectashield Hardset medium (Vector Laboratories). All images were collected using a Biorad Radiance 2100 confocal microscope with a 1.4NA 60× objective. Spindle angles were measured in ~20 cells for each condition and their distribution displayed in a radial histogram.

### Choanoflagellate culture, spindle orientation assays, and gene ortholog inference

Growth medium was prepared in artificial sea water. *S. rosetta* cultures (ATCC 50818) consisting primarily of chain colonies and slow swimmers were maintained by passaging 2 ml of culture into 18 ml fresh medium every day (*King et al., 2009*). Rosette colonies were produced by inoculating *S. rosetta* chain colonies with *Algoriphagus machipongonensis* bacteria (*Dayel et al., 2011*). Log phase *S. rosetta* cells were treated with 0.33 micromolar Nocodazole (Sigma M1404) for 18 hr at RT. Cells were pelleted by centrifugation for 5 min at 2000 x g, and washed thrice in artificial sea water to remove drug. Cells were allowed to recover for 30, 45, or 60 min at RT before fixation. Approximately 0.1 ml of cells were applied to poly-l-lysine-coated 96-well plates and left to attach for 30 min. Cells were fixed for 5 min with 0.2 ml 6% acetone, and then for 20 min with 0.2 ml 4% formaldehyde. Acetone and formaldehyde were diluted in artificial seawater, pH 8.0. Wells were washed gently four times with 1 ml washing buffer (100 mM PIPES at pH 6.9, 1 mM EGTA, and 0.1 mM MgSO$_4$) and incubated for 30 min in 1 ml blocking buffer (washing buffer with 1% BSA, 0.3% Triton X-100). Cells were incubated with primary antibodies diluted in 0.15 ml blocking buffer for 1 hr, washed four times with 0.2 ml of blocking buffer, and incubated for 1 hr in the dark with fluorescent secondary antibodies (1:1000 in blocking buffer, Alexa Fluor 488 goat anti-mouse, and Alexa Fluor 568 goat anti-rabbit; Invitrogen). Wells were washed thrice with washing buffer, blocked with 0.2 ml DAPI at 1 µg/ml for 5 min, and washed twice more. The following primary antibodies were used: Mouse monoclonal antibody against β-tubulin (E7, 1:100; Developmental Studies Hybridoma Bank) and nuclear pore complexes (1:100, Covance ab24609). Images were taken with a 63× oil immersion objective on a Leica DMI6000 B inverted compound microscope and Leica DFC350 FX camera.

The spindle angle for individual, non-colonial *S. rosetta* cells was measured from images of cells fixed and stained for tubulin and DNA as described above. Only mitotic cells with clear flagella, spindle poles, and cell body axis position were selected for analysis. We measured the acute angle between the spindle axis (defined as a line connecting the two spindle poles) and a line extending from the cell body's centroid to the midpoint between the duplicated flagella's basal bodies. The ImageJ software package was used to calculate the centroid and angle.

## Acknowledgements

We thank Aaron Turkewitz, Vincent Lynch, and Jamie Bridgham, Dave Anderson, and other members of the Thornton laboratory for comments on the manuscript. Supported by R01GM104397 (JWT), R01GM087457 (KEP), R01GM089977 (NK), and a Howard Hughes Medical Institute Early Career Scientist Award (JWT). NK is an investigator in the Howard Hughes Medical Institute and a Senior Scholar in the Integrated Microbial Biodiversity Program of the Canadian Institute for Advanced Research.

## Additional information

### Funding

| Funder | Grant reference number | Author |
|---|---|---|
| National Institute of General Medical Sciences | R01GM104397 | Joseph W Thornton |
| National Institute of General Medical Sciences | R01GM087457 | Kenneth E Prehoda |
| National Institute of General Medical Sciences | R01GM089977 | Nicole King |
| Howard Hughes Medical Institute | | Nicole King |

The funders had no role in study design, data collection and interpretation, or the decision to submit the work for publication.

### Author contributions

DPA, Conceived the project; designed, performed and interpreted phylogenetic, molecular, genomic, and structural analyses; contributed to writing the manuscript; DSW, Designed and interpreted molecular experiments; VHS, Designed, performed, and interpreted phylogenetic experiments; AW, Performed and interpreted microscopy of choanoflagellate cells; WCB, Performed molecular experiments; BFV, Designed molecular experiments; NK, Performed and interpreted microscopy of choanoflagellate cells; interpreted genomic analyses; contributed to writing the manuscript; JWT, Conceived the project; designed, performed, and interpreted molecular and structural analyses and microscopy of choanoflagellate cells; contributed to writing the manuscript; KEP, Conceived the project; designed, performed, and interpreted phylogenetic genomic, and structural analyses; designed and interpreted molecular analyses; wrote the manuscript

## Additional files

### Supplementary files

• Supplementary file 1. Alignment of GK domains used for phylogenetic analysis and reconstruction.

• Supplementary file 2. Phylogeny of GK domains in Newick format.

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
