## [Decision Letter]

Thank you for submitting your work entitled "Evolution of a new protein function required for organized multicellularity in animals" for peer review at *eLife*. Your manuscript has been evaluated by Detlef Weigel as Senior editor, Jesse Bloom as Reviewing editor, as well as by two anonymous reviewers with substantial expertise in spindle positioning.

The reviewers have discussed the reviews with one another and the Reviewing editor has drafted this decision to help you prepare a revised submission.

We find considerable value in your work. It is remarkable that one or two ancestral mutations are sufficient to convert GK_PID_ from a guanylate kinase to a protein capable of binding PINS in a species (e.g. *Drosophila*) greatly diverged from this ancestor. Your biophysical characterization of this reconstructed protein and your assessment of this characterization to uncertainty in the reconstruction are major pluses. This study represents a notable advance over most existing studies of protein evolution since it examines the wholesale repurposing of a protein rather than an incremental shift in activity or specificity.

We also appreciate the innovative mix of approaches that you have used to elucidate some of the steps towards the evolution of organized multicellularity. This is a complex and multi-factorial evolutionary transition, and a better understanding of its basis is of great interest.

Our major critique is that the broader interpretation is overstated in terms of the centrality of KHC73-DLG-PINS to spindle orientation in all animals and multicellularity in general, and in terms of external orientation being a unique novelty of animals. We also think that some clarification / caveats are needed regarding the experiments on positioning in Choanoflagellates. Finally, we think the manuscript would benefit from more discussion of some puzzling aspects of the co-evolution of GK_PID_ and PINS (or lack thereof).

*eLife* attempts to provide reviews that do not require new work beyond the scope of the original manuscript. It is our opinion that the critiques can be addressed largely or completely by textual changes – even if your work leaves many questions about the evolution of spindle orientation unanswered, it is a sufficient step forward to be of interest to the *eLife* readership. However, please take seriously the task of making revisions that address the critiques below. Depending on the revisions, I may be able to editorially evaluate a revised manuscript myself, or I may have to consult with reviewers who have more expertise in spindle orientation.

Major critiques:

1) The implication that all oriented cell division in animals is due to GK_PID_ is overstated. Nearly all of the data and cited literature is specific to *Drosophila*. For instance, in Figure 3 it appears that the PINS of Worm (presumably *C. elegans*?) doesn't even have the phosphoserine posited to be recognized by GK_PID_, although worms obviously orient their spindles. Other pathways have been implicated in oriented cell division. One reviewer notes that, "cell division in early embryos follows Hertwig's rule (i.e. that cells tend to divide along their long access) and it has been argued that Myosin activity is the primary determinant of the division plane in *C. elegans* neuroblasts. I am unaware of evidence that those two phenomena are primarily caused by GK_PID_." Another reviewer notes that, "Even in the *Drosophila* S2 cell induced cell polarity assay, the function of this complex is mostly studied in the context of TPR-deleted PINS. TPR-deleted PINS cannot bind Mud (NuMA). The PINS/MUD pathway IS highly conserved in multiple phyla. I would be happy if the authors could cite high quality data for a spindle positioning role for KHC73:DLG:PINS in ONE additional phylum other than Arthropoda. There are many papers on the mammalian scribbled polarity complex. Polarity is not the same thing as spindle orientation." We do not expect you to undertake new experiments to test the essentiality of GK_PID_ for oriented cell division across all animals, but it is absolutely essential to make substantial textual changes to reflect the fact that although current evidence indicates that GK_PID_ is an important mechanism for spindle orientation in some organisms, neither your data nor the literature currently support it being a universal mechanism in metazoans.

2) The claims about the flagellar basal body serving as the microtubule organizing center in Choanoflagellates seem overstated, particularly as this is a substantially new claim. First, this section should include data about the number of cells analyzed. Second, the following statement seems to be more of a hypothesis than a fully supported claim: "During mitosis, the flagellar basal body duplicates; the daughter bodies migrate symmetrically away from the cell's basal pole and then serve as microtubule organizing centers for assembling the spindle and orienting it perpendicular to the A-B axis, a process that may or may not involve GK_PID_. Because this mode of spindle orientation relative to the cellular axis occurs in both colonial and non-colonial *S. rosetta* cells, it must involve internal marks imposed by the cell's polarity rather than cues from neighboring cells." Unless you are prepared to undertake substantial new experiments to fully demonstrate that the basal bodies are in fact the organizing centers, this statement and similar ones should be dramatically tempered.

3) Your work clearly demonstrates that GK_PID_ evolved the latent capacity to bind PINS long before it appears to have actually been paired with a PINS that it could bind. We do not think this is an artifact, as your ancestral reconstruction samples broadly across species and the result is robust to the reconstruction – so this somewhat puzzling finding does appear to be true. It may be impossible to explain exactly why this occurred, but more discussion of this conundrum is warranted. Why should the ancestral and Choanoflagellate GK_PID_ bind a *Drosophila* PINS but not the PINS in that same organism? This question is going to come up in the mind of every reader, so your best guesses at plausible explanations would be helpful.

Minor comments:

The authors write: "[…] for the evolution of a new protein function crucial to the emergence of animal multicellularity." While GK_PID_ is certainly very important in at least some modern animals, that does not mean that it is important for the "emergence" of animal multicellularity. This statement should be clarified, qualified, or removed.

The author's write: "[…] Spindle orientation itself is not a metazoan novelty; indeed, most eukaryotes orient the mitotic spindle, but most appear to do so relative to the cell's internal structure." It's also true that in many metazoan cell divisions the spindle is responding to "the cell's internal structure". Indeed, in many contexts that internal structure may be the distribution of GK_PID_ proteins. Of course, that distribution may be controlled by external signals.

The authors write: "Orientation of the mitotic spindle relative to an externally induced molecular mark was an essential novelty in the evolution of animals and the emergence of a higher level of biological organization – the multicellular individual". What is their evidence that there are no single-celled Eukaryotes that orient their spindle relative to external cues? One could certainly imagine some single-cell organisms might orient their spindle based on environmental cues. The authors should remove or better justify their statement that this is "an essential novelty".

It might be helpful to include a brief discussion of the orientation of cell division in plants, which are multicellular organisms that do not use GK_PID_.

The actual sequence of the PINS peptide used in anisotropy assays is not given anywhere; how does it relate to those in Figure 3? This is actually very important to clarify.

The amino acid sequences of the GK domains used in biochemical experiments should be given.

Since all the GK proteins were purified by gel filtration, showing the purity and gel filtration profiles of all the GK domains purified as Figure supplements would be appropriate. Aggregated protein would be one artifactual reason for binding PINS linker peptide in anisotropy assays while having no guanylyl kinase activity.

A clearer description of the reconstruction of DLGs used in the S2 cell assay would be helpful.

Is it really necessary to introduce the acronym of APR for ancestral protein reconstruction? There are already a lot of gene name acronyms.

Does the choanoflagellate PINS have the putative phospho-serine 436?

---

## [Author Response]

*Major critiques:*

*1) The implication that all oriented cell division in animals is due to GK_PID_ is overstated. Nearly all of the data and cited literature is specific to* Drosophila. *For instance, in Figure 3 it appears that the PINS of Worm (presumably* C. elegans*?) doesn't even have the phosphoserine posited to be recognized by GK_PID_, although worms obviously orient their spindles. Other pathways have been implicated in oriented cell division. One reviewer notes that, "cell division in early embryos follows Hertwig's rule (i.e. that cells tend to divide along their long access) and it has been argued that Myosin activity is the primary determinant of the division plane in* C. elegans *neuroblasts. I am unaware of evidence that those two phenomena are primarily caused by GK_PID_." Another reviewer notes that, "Even in the* Drosophila *S2 cell induced cell polarity assay, the function of this complex is mostly studied in the context of TPR-deleted PINS. TPR-deleted PINS cannot bind Mud (NuMA). The PINS/MUD pathway IS highly conserved in multiple phyla. I would be happy if the authors could cite high quality data for a spindle positioning role for KHC73:DLG:PINS in ONE additional phylum other than Arthropoda. There are many papers on the mammalian scribbled polarity complex. Polarity is not the same thing as spindle orientation." We do not expect you to undertake new experiments to test the essentiality of GK_PID_ for oriented cell division across all animals, but it is absolutely essential to make substantial textual changes to reflect the fact that although current evidence indicates that GK_PID_ is an important mechanism for spindle orientation in some organisms, neither your data nor the literature currently support it being a universal mechanism in metazoans.*

We take this point seriously, because studies of the mechanisms of spindle orientation only scratch the surface of the diversity of animal taxa and cell types within them. We have modified the text in numerous ways to be more cautious on this point and to base our claims more solidly on what is known in the literature. Recent work indicates that the GK_PID_ complex coordinates spindle orientation not only in *Drosophila* neuroblasts but also in both mammals and birds and in other cell types, including epithelia from numerous tissues; we have added discussion of these citations to the manuscript. The role of the GK_PID_ complex in spindle orientation in both protostomes and deuterostomes and in multiple cell types suggests an ancient and general role. However, this does not mean that the role of GK_PID_ is the universal mediator of spindle orientation in all animal cells: other mechanisms may be important in certain taxonomic lineages and cell types, and we have changed the text to explicitly address these points. We have also discussed the role of other molecules and pathways in spindle orientation.

More generally, we have gone through the text and have changed our wording to dispel the impression that GK_PID_ complex is the sole driver of spindle orientation in all animals and all cell types and to avoid the implication that the evolution of the GK_PID_ complex explains all instances of spindle orientation in all animals.

*2) The claims about the flagellar basal body serving as the microtubule organizing center in Choanoflagellates seem overstated, particularly as this is a substantially new claim. First, this section should include data about the number of cells analyzed. Second, the following statement seems to be more of a hypothesis than a fully supported claim: "During mitosis, the flagellar basal body duplicates; the daughter bodies migrate symmetrically away from the cell's basal pole and then serve as microtubule organizing centers for assembling the spindle and orienting it perpendicular to the A-B axis, a process that may or may not involve GK_PID_. Because this mode of spindle orientation relative to the cellular axis occurs in both colonial and non-colonial* S. rosetta *cells, it must involve internal marks imposed by the cell's polarity rather than cues from neighboring cells." Unless you are prepared to undertake substantial new experiments to fully demonstrate that the basal bodies are in fact the organizing centers, this statement and similar ones should be dramatically tempered.*

We believe that the role of the basal bodies acting as microtubule organizing centers in mitosis is well established in the literature. According to the description of mitosis in Leadbeater’s canonical book in the field, "The flagellar and non-flagellar basal bodies separate, the latter moving towards the cell surface just beneath the plasma membrane. At this stage both basal bodies possess a ring of microtubules […] The two basal bodies move apart and become foci (poles) for the developing mitotic spindle.” (Leadbeater, p. 40, 2015). Leo Buss’ book The Evolution of Individuality also discusses the evidence for this mechanism in detail.

We have added these references to support this point. We have also softened the language claim to make clear that the scenario we propose for the ancient mode of spindle orientation is a hypothesis to be further tested. We also reorganized this section of the text to improve its clarity. As requested, we also included the number of choanoflagellate cells analyzed for both colonial and solitary spindle orientation (12 and 7, respectively).

*3) Your work clearly demonstrates that GK_PID_ evolved the latent capacity to bind PINS long before it appears to have actually been paired with a PINS that it could bind. We do not think this is an artifact, as your ancestral reconstruction samples broadly across species and the result is robust to the reconstruction – so this somewhat puzzling finding does appear to be true. It may be impossible to explain exactly why this occurred, but more discussion of this conundrum is warranted. Why should the ancestral and Choanoflagellate GK_PID_ bind a* Drosophila *PINS but not the PINS in that same organism? This question is going to come up in the mind of every reader, so your best guesses at plausible explanations would be helpful.*

We agree that this is puzzling. We have addressed this point briefly in the text, acknowledging the surprising nature of the result and suggesting the possibility is that the surface of GK-_PID_ that fortuitously binds *Drosophila* Pins might be used to bind another structurally similar ligand, possibly an ancient one. Because whatever we say here would be very speculative, we did not go into much detail on this point.

*Minor comments: The authors write: "[…] for the evolution of a new protein function crucial to the emergence of animal multicellularity." While GK_PID_ is certainly very important in at least some modern animals, that does not mean that it is important for the "emergence" of animal multicellularity. This statement should be clarified, qualified, or removed.*

We agree and have rewritten this paragraph.

*The author's write: "[…] Spindle orientation itself is not a metazoan novelty; indeed, most eukaryotes orient the mitotic spindle, but most appear to do so relative to the cell's internal structure." It's also true that in many metazoan cell divisions the spindle is responding to "the cell's internal structure". Indeed, in many contexts that internal structure may be the distribution of GK_PID_ proteins. Of course, that distribution may be controlled by external signals.*

This is certainly true. We have attempted to be more precise by revising our statement to read: "Spindle orientation itself is not a metazoan novelty; indeed, most eukaryotes – even unicellular ones – orient the mitotic spindle, but most appear to do so relative to the cell’s internal structure alone rather than in response to cues from adjacent cells."

*The authors write: "Orientation of the mitotic spindle relative to an externally induced molecular mark was an essential novelty in the evolution of animals and the emergence of a higher level of biological organization – the multicellular individual". What is their evidence that there are no single-celled Eukaryotes that orient their spindle relative to external cues? One could certainly imagine some single-cell organisms might orient their spindle based on environmental cues. The authors should remove or better justify their statement that this is "an essential novelty".*

We have removed the sentence to which the reviewer objected and have emphasized instead that cues from adjacent cells per se – rather than external cues more generally – are important for tissue organization in animals. We have also avoided calling this phenomenon a novelty.

*It might be helpful to include a brief discussion of the orientation of cell division in plants, which are multicellular organisms that do not use GK_PID_.*

We have changed the text to acknowledge that multicellularity evolved repeatedly in different lineages and that key cellular functions typically have different underlying mechanisms. We did not go into detail about the specific mechanisms of spindle orientation in plants, in the interest of keeping the paper concise.

*The actual sequence of the PINS peptide used in anisotropy assays is not given anywhere; how does it relate to those in Figure 3? This is actually very important to clarify.*

We have provided this sequence in the Methods section, and we say explicitly that it is the peptide from *D. melanogaster*.

*The amino acid sequences of the GK domains used in biochemical experiments should be given.*

These sequences have been deposited in GenBank; we added the identifiers to manuscript in the Methods section. The sequences are also found in Figure 1—figure supplement 1 and Figure 1—figure supplement 2.

*Since all the GK proteins were purified by gel filtration, showing the purity and gel filtration profiles of all the GK domains purified as Figure supplements would be appropriate. Aggregated protein would be one artifactual reason for binding PINS linker peptide in anisotropy assays while having no guanylyl kinase activity.*

As noted in the Methods section, all the GK proteins eluted in size exclusion chromatography at volumes consistent with the molecular weight of a monomer. Unfortunately, we did not save the chromatograms from the gel filtration runs to include them in the supplementary material.

*A clearer description of the reconstruction of DLG's used in the S2 cell assay would be helpful.*

We have clarified which Dlg constructs were used in the S2 cell assay in the figure legend of the revised manuscript and in the Methods. The sequences of the GKPIDs used were the same as those used in the other assays; we have clarified this point.

*Is it really necessary to introduce the acronym of APR for ancestral protein reconstruction? There are already a lot of gene name acronyms.*

We agree and have spelled the phrase out.

Does the choanoflagellate PINS have the putative phospho-serine 436?

No, it does not. This is included in Figure 3.